

# Comprehensive identification and expression analysis of *FAR1/FHY3* genes under drought stress in maize (*Zea mays* L.)

Dongbo Zhao[1], Peiyan Guan[2], Longxue Wei[1], Jiansheng Gao[1], Lianghai Guo[1], Dianbin Tian[3], Qingfang Li[4], Zhihui Guo[1], Huini Cui[1], Yongjun Li[1] and Jianjun Guo[1]

[1] Dezhou Academy of Agricultural Science, Dezhou, Shandong, China
[2] College of Life Science, Dezhou University, Dezhou, Shandong, China
[3] Pingyuan County Rural Revitalization Service Center, Pingyuan, Shandong, China
[4] Linyi County Agricultural and Rural Bureau, Linyi, Shandong, China

Corresponding author
Jianjun Guo, gjj3185@163.com

## ABSTRACT

**Background:** FAR1/FHY3 transcription factors are derived from transposase, which play important roles in light signal transduction, growth and development, and response to stress by regulating downstream gene expression. Although many FAR1/FHY3 members have been identified in various species, the *FAR1/FHY3* genes in maize are not well characterized and their function in drought are unknown.
**Method:** The FAR1/FHY3 family in the maize genome was identified using PlantTFDB, Pfam, Smart, and NCBI-CDD websites. In order to investigate the evolution and functions of FAR1 genes in maize, the information of protein sequences, chromosome localization, subcellular localization, conserved motifs, evolutionary relationships and tissue expression patterns were analyzed by bioinformatics, and the expression patterns under drought stress were detected by quantitative real-time polymerase chain reaction (qRT-PCR).
**Results:** A total of 24 ZmFAR members in maize genome, which can be divided into five subfamilies, with large differences in protein and gene structures among subfamilies. The promoter regions of *ZmFARs* contain abundant abiotic stress-responsive and hormone-respovensive *cis*-elements. Among them, drought-responsive *cis*-elements are quite abundant. *ZmFARs* were expressed in all tissues detected, but the expression level varies widely. The expression of *ZmFARs* were mostly down-regulated in primary roots, seminal roots, lateral roots, and mesocotyls under water deficit. Most *ZmFARs* were down-regulated in root after PEG-simulated drought stress.
**Conclusions:** We performed a genome-wide and systematic identification of *FAR1/FHY3* genes in maize. And most *ZmFARs* were down-regulated in root after drought stress. These results indicate that FAR1/FHY3 transcription factors have important roles in drought stress response, which can lay a foundation for further analysis of the functions of *ZmFARs* in response to drought stress.

## INTRODUCTION

As sessile organisms, plants are frequently challenged by abiotic stresses, such as drought, salt, temperature, heavy metal ions and ultraviolet radiation, which severely affect their growth and reduce their yield (*Zhang et al., 2022*). Meanwhile, plants are autotrophic organisms and light is the basis of photosynthesis, therefore, studying the expression patterns and functions of light-related genes in plants can improve yield-related agronomic traits. FHY3 (far-red elongated hypocotyls 3) and FAR1 (far-red-impaired response) are two homologous proteins derived from transposases, which were initially identified in Arabidopsis, as an important component of the phytochrome A (phyA)-mediated far-red light signaling pathway (*Hudson et al., 1999*; *Wang & Deng, 2002*). FHY3 and FAR1 are plant-specific transcription factors, with separable DNA-binding domain, namely N-terminal C2H2 zinc finger domain, and transcriptional activation domains, including a central putative core transposase domain and a C-terminal SWIM motif (named after SWI2/SNF and MuDR transposases), which are essential for forming homo- or heterodimers to regulate and modulate downstream gene expression (*Lin et al., 2008*; *Makarova, Aravind & Koonin, 2002*). Studies have shown that FHY3 and FAR1 have multifaceted roles in light signaling, physiological and developmental processes, as well as in response to abiotic stresses (*Lin et al., 2007*; *Ma & Li, 2018*; *Tang et al., 2013*; *Zheng, Sun & Liu, 2023*).

In Arabidopsis, *FHY3* and *FAR1* encode two proteins related to mutator-like transposases that co-regulate phyA nuclear accumulation, and participate in the plant light signalling response by activating the transcription of *FHY1* and *FHL* through direct binding to the FBS motif "CACGCGC" in the promoter (*Lin et al., 2007*). *FHY3* and *FAR1* act as positive regulators of *ELF4* and *CCA1*, which are involved in the regulation of plant biological rhythms (*Li et al., 2011*; *Liu et al., 2020*). *FHY3* is the epistatic regulator of WUSCHEL (WUS) and CLAVAT3 (CLV3), two central players in the establishment and maintenance of meristems, which in turn regulate flowering time (*Li et al., 2016*). Besides, FHY3 binds directly to the promoter and activates the expression of *ACCUMULATION AND REPLICATION OF CHLOROPLASTS5 (ARC5)* and *HEMB1*, which regulates chloroplast development and chlorophyll biosynthesis, respectively (*Ouyang et al., 2011*; *Tang et al., 2012*). FHY3 negatively regulate age- and light-mediated leaf senescence by repressing the transcription of *WRKY28* (*Tian et al., 2020*). In addition, FHY3/FAR1 binds to the promoter of *MIPS1* to activate its expression directly, thereby promoting inositol biosynthesis to prevent light-induced oxidative stress and SA-dependent cell death (*Ma et al., 2016*). Moreover, FHY3 and FAR1 promotes branching and stress tolerance in *Arabidopsis thaliana* by integrating auxin and strigolactone signalling (*Liu et al., 2020*; *Stirnberg et al., 2012*). Furthermore, FHY3 and FAR1 can also responds to ABA signaling to regulate seed germination, seedling development, and primary root growth (*Tang et al., 2013*), and through the transcriptional activation of starch-debranching enzyme ISOAMYLASE2 (ISA2) affects starch synthesis and starch granule formation (*Ma et al., 2017*).

FHY3/FAR1 family has been identified in many plant species, including Arabidopsis, tea, cucumber, peanut, walnut, and potato (*Chen et al., 2023b*, *2023c*; *Li et al., 2023*; *Lin & Wang, 2004*; *Liu et al., 2021*; *Lu et al., 2022*). FHY3/FAR1 family has been reported to be associated with plant development in Arabidopsis, peanut, and walnut. However, in tea, potato and cucumber the researchers were mainly focused on stresses. The role of *FAR1* in abiotic stresses has attracted attention in recent years. In tea, *CsFHY3/FAR1s* were strongly expressed in leaves, and the expression of most genes were induced under salt stress, and negatively expressed under low temperature stress (*Liu et al., 2021*). In *Arachis hypogaea*, overexpression of the *AhJ11-FAR1-5* can enhance tolerance to drought stress by increasing POD, SOD, and CAT scavenging (*Yan et al., 2020*). In Arabidopsis, *fhy3* and *far1* mutants are less sensitive to salt, osmotic, while more sensitive to drought than the wild type (*Tang et al., 2013*). In rice, FHY3/FAR1 family member *TSD1* is induced by heat and highly expressed in spikelets, and specifically enhances its thermotolerance during spikelet morphogenesis (*Cai et al., 2023*). In potato, most *StFRS* genes were down-regulated by low temperature and polyethyleneglycol (PEG) treatment (*Chen et al., 2023b*). These results suggested that FHY3/FAR1 genes play important roles in the response to abiotic stress.

Maize (*Zea mays* L.) is one of the most important cereal crops in the world, which is grown over a very wide area, between 58°N and 40°S latitude. Maize seedlings are sensitive to drought stress, especially in the early growth stage, and drought severely influence maize production. Plant roots and mesocotyl play important roles in sensing environmental water stress (*Saenz Rodriguez & Cassab, 2021*). Maize has a complex root system that consists of primary, secondary and aerial roots. The formation of these root types is characterized by temporal and spatial developmental variability, implying that they have specific functions during maize development (*de Dorlodot et al., 2007*). At the early seedling stage, roots and mesocotyls can respond sensitively to phytohormones and environmental stresses, and they are important organs as evaluators of stress tolerance (*Niu et al., 2020*; *Zhang et al., 2023*). Previous studies have shown that *FHY3/FAR1* genes participate in response to drought stress, but their functions in maize are still unknown. It is valuable to identify the FHY3/FAR1 family members, and to clarify their functions under drought stress. In this study, we used bioinformatics analyzed the number and classification, gene structure, chromosomal localization, and tissue expression patterns of the maize FAR1/FHY3 family members. In addition, qRT-PCR was used to detect the expression patterns of the *ZmFAR1s* in the roots of B73 seedlings at one-leaf and three-leaf stage under drought stress, revealing the molecular characteristics of the FAR1/FHY3 genes in maize. This study will provide the basis for further research on the biological functions of the FAR1/FHY3 gene family in maize, and also has important reference value for the genetic improvement of drought-tolerant maize lines.

## MATERIALS AND METHODS

### Plant materials and treatments

The seeds of maize inbred line B73, were cultivated in a growth chamber under long day conditions (16 h of light, 25 °C, and 8 h of darkness, 22 °C). In order to study the

expression patterns of *ZmFHY3/FAR1* in roots and mesocotyls under drought stress, the control was watered normally, while the experimental group without watering. The roots and mesocotyls were harvested at one-leaf (V1) stage (*Tai et al., 2016*). For study the expression patterns of *ZmFHY3/FAR1* at three-leaf (V3) stage in root, seedlings were treated with 10% Polyethylene glycol (PEG)-6000 (w/v) and 25% PEG-6000 aqueous solution (w/v) for 6, 24 and 48 h, respectively. Six seedlings were treated per sample, and three biological replicates were conducted for each sample. The sampling parts of maize seedlings are shown in Fig. S1. All samples were frozen in liquid nitrogen and stored at −80 °C for later use.

## Identification and evolutionary analysis of *FAR1/FHY3* gene family members

FAR1/FHY3 protein sequences of maize were obtained from Plant TFDB (https://planttfdb.gao-lab.org/). Using the Pfam (http://pfam.xfam.org/), Smart (http://smart.embl.de/), and NCBI online tools Conservative Domain Database (CDD) (https://www.ncbi.nlm.nih.gov/cdd/) to verify FAR/FHY3 members. Protein sequence comparison was performed with MegAlign software in DNASTAR package (Lasergene, Madison, WI, USA). Protein sequences of sugarcane, and Arabidopsis were also downloaded, and phylogenetic trees were constructed using the neighbor-joining method (NJ, bootstrap = 1,000) in MEGA 7.0 software (*Kumar et al., 2018*). The collinearity relationship between different gene pairs was performed using MCScanX (*Wang et al., 2012*). Finally, the results were visualized using the Dual Systeny Plot for MCScanX package in the TBtools (*Chen et al., 2023a*).

## The characterization of ZmFAR proteins and subcellular localization analysis

The .gff file was downloaded from Maize GDB (https://maizegdb.org/), which can query the chromosome location and structure information of FAR1/FHY3. Based on the localization in chromosome, the *ZmFAR1/FHY3* genes were renamed. Protein characterization information was obtained from the ProtParam (https://web.expasy.org/protparam/) tool in Expasy. Plant-mPLoc (http://www.csbio.sjtu.edu.cn/) was used for subcellular localization analysis, and SOMPA (Institut de Biologie et Chimie de Proteines, Lyon, France), was used to predict ZmFARs protein secondary structures.

## Conserved motifs and promoter analysis of *ZmFAR*s

Motif analysis was performed using the online tool MEME (https://meme-suite.org/meme/). Firstly, the promoter sequences (2,000 bp upstream of CDS) were obtained using the .gff3 sequence extraction tool in TBtool, and then submitted to the PlantCARE website (http://bioinformatics.psb.ugent.be/) for *cis*-acting element scan. And integrated for mapping by using the Gene Structure View tool in TBtools software (*Chen et al., 2023a*).

### Expression patterns analysis of *ZmFARs*

The transcriptome dataset of maize genes was downloaded from the NCBI database (GSE50191) (*Walley et al., 2016*) and visualized using TBtools software, for tissues expression patterns analysis.

Total RNA was extracted using RNAiso Plus (TaKaRa, Shiga, Japan). The concentration and purity of nucleic acids are determined by NanoDrop2000. cDNA was obtained by reverse transcription reaction using PrimeScript™ RT reagent Kit with gDNA Eraser (TaKaRa, Shiga, Japan). The cDNA template was diluted for 30-fold and then stored at −20 °C for later use. *ZmFAR*-specific primers were designed using Beacon Designer software (Table S1), and the expression level of *ZmFARs* was detected by CFX96 PCR instrument (Bio-Rad, Hercules, CA, USA). The qRT-PCR reaction system (15 μL) consisted of 7.5 μL of 2 × TB Green Premix Ex Taq™ II (TaKaRa, Shiga, Japan), 0.45 μL upstream and downstream specific primers, 1.6 μL of ddH$_2$O and 5 μL of cDNA template. The reaction program was 95 °C pre-denaturation for 30 s; 95 °C denaturation for 5 s, 60 °C for 30 s, 72 °C extension for 10 s, and 40 cycles. Melt curve 65 °C to 95 °C, increment 0.5 °C. Three replications were performed for each sample, and the corresponding Ct values were obtained for different samples. After homogenization of the internal reference gene *Actin 1*, the relative expression of genes was calculated by the $2^{-\Delta\Delta Ct}$ method (*Livak & Schmittgen, 2001*). Finally, the data is visualized using GraphPad prism.

## RESULTS

### Screening and identification of ZmFAR family members

A total of 24 putative members with typical FAR1/FHY3 structural domains were identified from maize genome, and they were encoded by 14 genes (Fig. 1). Their CDS and protein sequences are listed in Table S2. They all have N-terminal WRKY-GCM1 zinc finger domain with the conserved cysteines and histidines of the CCHH motif (Fig. S2). And a putative "DDE" catalytic triad motif (E323 is not conserved, while G305 is conserved in FHY3) that is critical for transposase/integrase function, and C-terminal SWIM zinc-finger domain of a CxCxnCXH motif were found in all ZmFARs, expect ZmFAR01.1 (Fig. S2). Based on the chromosomal distribution, they were renamed ZmFAR01.1 to ZmFAR14.1. And they were unevenly distributed on seven chromosomes of maize, mainly on Chr1, Chr5, and Chr7. Among them, Chr7 has the highest distribution with 9 members. However, no ZmFAR was present in Chr2, Chr6, and Chr8 (Fig. 1). According to the CDD website, all members contained FHY3 conserved structural domains, among which eight members contained 1–2 FAR1 structural domains, while the rest of them did not contain FAR1 domain (Fig. 2C). These results indicated that the identification of 24 FHY3/FAR1 family members in maize was accurate.

### Phylogenetic analysis of the ZmFARs

In order to understand the evolutionary history and phylogenetic relationship of the FAR1/FHY3 genes family, 54 FAR1/FHY3 protein sequences (24 for maize, four for

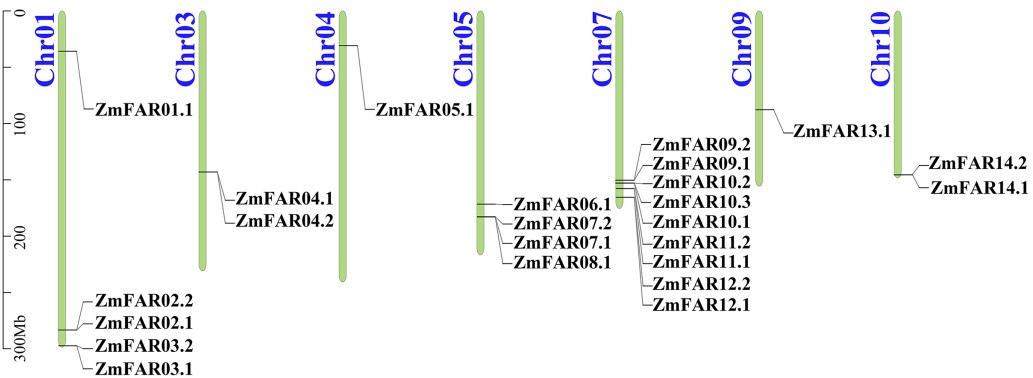

**Figure 1  The location of ZmFAR family members on chromosome.** Chromosome numbers are on the left and ZmFARs are on the right of chromosomes. Scale bar on the left indicates chromosome length.

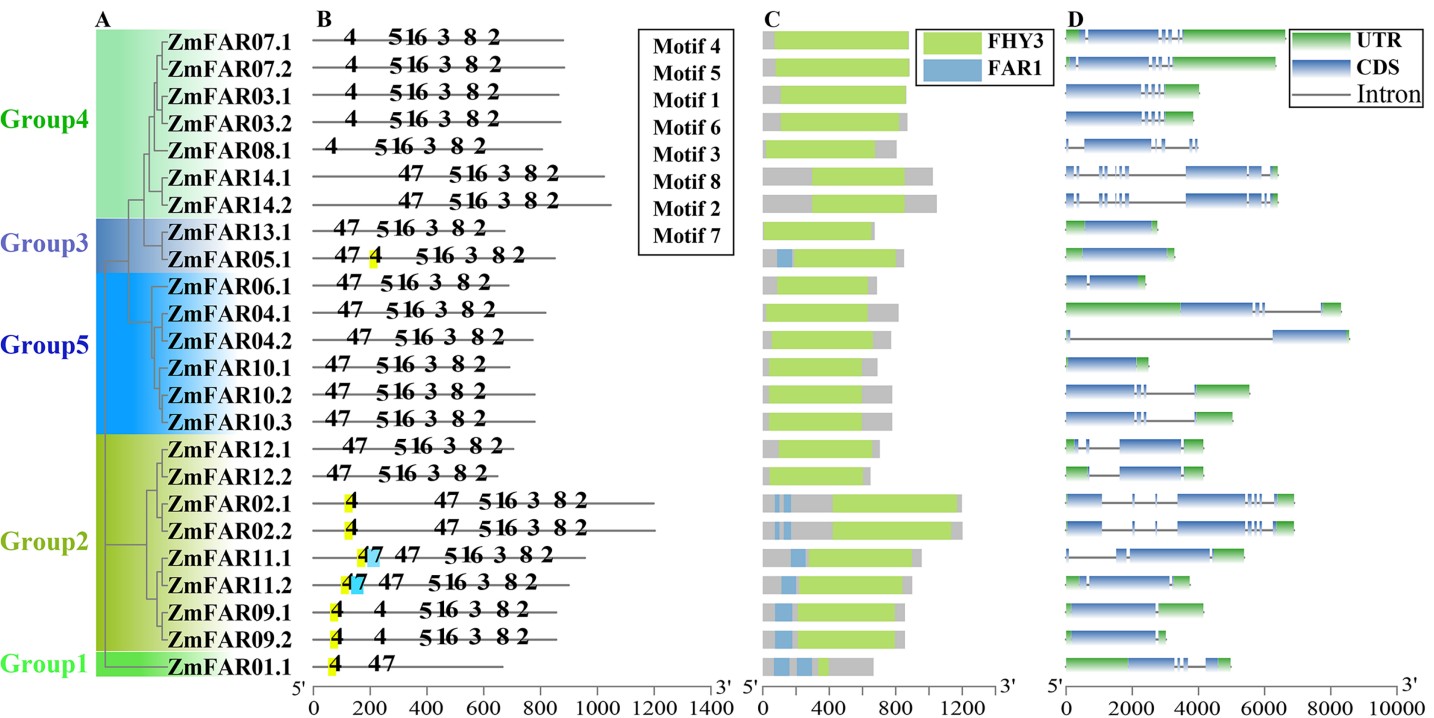

**Figure 2  The phylogenetic, conservative motifs and gene structure analysis of *ZmFARs*.** (A) The unrooted phylogenetic tree constructed using ZmFAR protein sequences. The subtree branch lines are colored indicate the different subfamilies. (B) Motifs analysis of ZmFARs. The top eight motifs identified by the MEME are represented by the number. (C) Conservative domains analyzed by NCBI-CDD. (D) Exon-intron structures of ZmFARs. The exons are marked as blue boxes, and the introns are represented by black lines; UTRs are shown as green boxes.

sugarcane, and 26 for Arabidopsis) were compared, and a phylogenetic tree was constructed using MEGA 7.0 software. The results showed that the FAR1/FHY3 can be divided into six subfamilies based on their sequence similarity (Fig. 3). The ZmFAR family members were most distributed in three subfamilies, mainly Group2, Group4 and Group5, with a total of 21 members; Group1 and Group3 had three members (ZmFAR01.1,

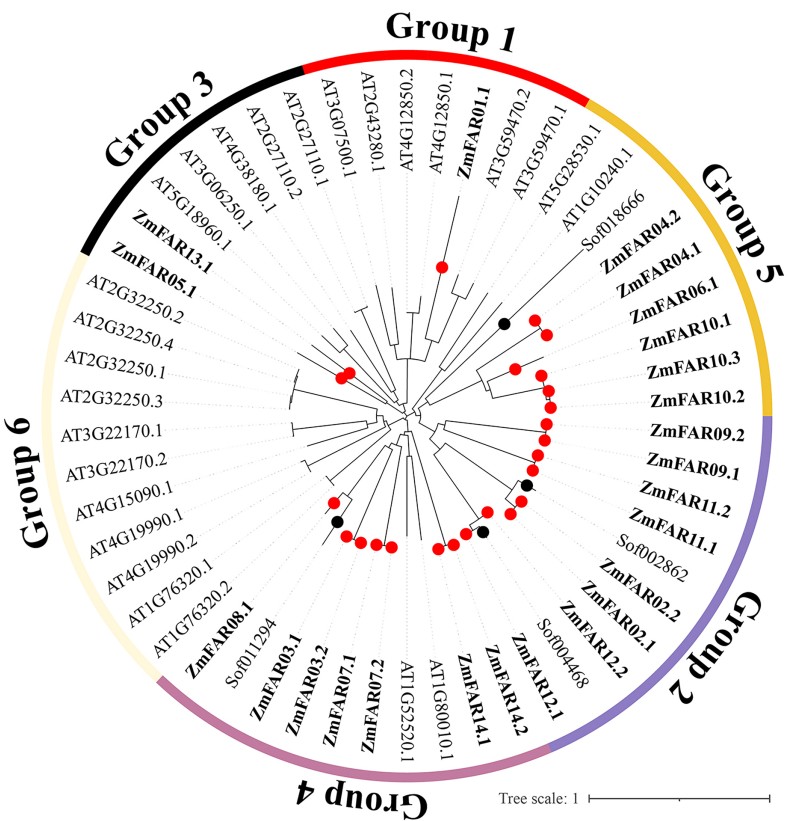

**Figure 3 Evolutionary relationships of FAR1/FHY3 transcription factors in maize, sugarcane, and Arabidopsis.** Zm indicates *Zea maize* (red dots), Sof indicates *Saccharum officinarum* (black dots), At indicates *Arabidopsis thaliana*. These FAR1/FHY3 proteins were divided into six groups, and were represented by different colors, respectively.

ZmFAR05.1 and ZmFAR13.1); In contrast, Group6 had no ZmFAR family members. Notably, *FAR1/FHY3* genes from maize and sugarcane showed close pairwise relationships based on genetic distance, compared with other proteins from different species, implying that the relationship between maize and sugarcane is closer than that between maize and other species.

## Collinearity analysis of ZmFARs

Collinearity analysis can elucidate the evolutionary history of genomes and gene families (*Wang et al., 2012*). To investigate the molecular mechanism of ZmFARs evolution, we analyzed the co-linearity of ZmFAR members among maize and other species by using MCScanX. Five monocots (*Sorghum bicolor*, *Oryza sativa*, *Oryza indica*, *Saccharum spontaneum*, and *Hordeum vulgare*), and four dicots (*Arabidopsis thaliana*, *Solanum tuberosum*, *Solanum lycopersicum*, and *Glycine max*) were applied for the co-linearity analyses. Surprisingly, there was no orthologous genes were identified in maize (Fig. S3), and dicots (Fig. S4). However, a total of 14, 12, 11, 16, and eight FAR1 paralogous gene pairs were identified in *Sorghum bicolor*, *Oryza sativa*, *Oryza indica*, *Saccharum spontaneum*, and *Hordeum vulgare*, respectively (Fig. 4 and Table S3). Notably, six ZmFAR

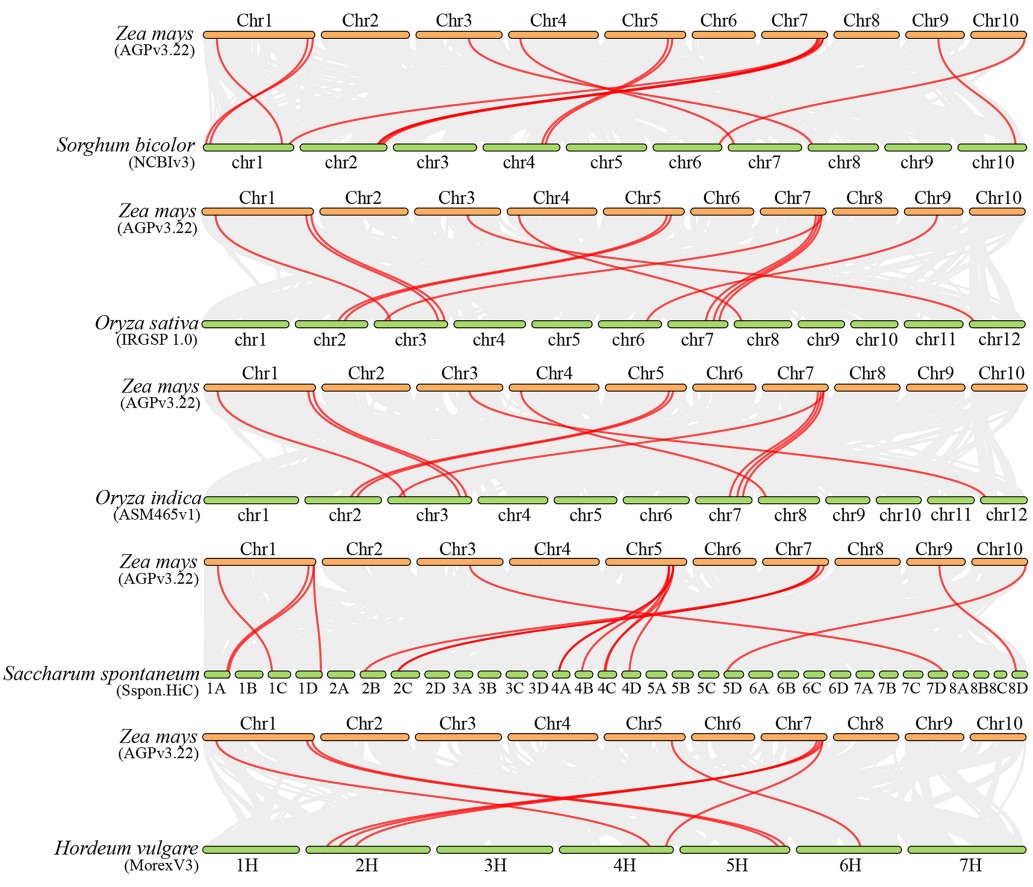

**Figure 4 Synteny analysis of ZmFARs between maize and other five monocots, including *Sorghum bicolor, Oryza sativa, Oryza indica, Saccharum spontaneum,* and *Hordeum vulgare.*** Gray lines represent the collinear blocks between two plants in their genome. Syntenic *FAR1/FHY3* gene pairs are represented by the marked red lines.

members (ZmFAR03.2, ZmFAR02.2, ZmFAR01.1, ZmFAR07.2, ZmFAR12.1, and ZmFAR09.2) had syntenic pairs throughout all five monocots, which were further indicated in bold in Table S4. While, ZmFAR08.1 had one syntenic pair with sugarcane, indicating that they have a common ancestor (*Abrouk et al., 2010*). These results suggest that FAR1 genes evolved after the differentiation of monocots and dicots plants.

## Protein characterization and subcellular localization analysis of the ZmFARs

The individual characteristics- including their physiological and biochemical properties, and cellular location are summarized in Table 1. The protein sequences and physicochemical properties of different ZmFAR transcription factors varied greatly, with amino acid lengths ranged from 648aa (ZmFAR12.2) to 1202aa (ZmFAR02.2), molecular weights ranged from 73.27 kDa (ZmFAR01.1) to 135.849 kDa (ZmFAR02.2), theoretical isoelectric points ranged from 5.31 (ZmFAR04.1) to 8.99 (ZmFAR08.1). There are seventeen basic proteins and seven acidic proteins, respectively. The grand average of hydropathicity of all ZmFAR proteins were less than 0, indicating that they all belonged to

**Table 1 Basic information of maize FAR1/FHY3 transcription factor family members.**

| ID | Gene name | Number of amino acids | Molecular weight | Theoretical pI | Instability index | Grand average of hydropathicity | Subcellular localization |
|---|---|---|---|---|---|---|---|
| ZmFAR01.1 | GRMZM2G001663 | 666 | 73,270.02 | 7.83 | 53.37 | −0.477 | Nucleus |
| ZmFAR02.1 | GRMZM2G463730 | 1,198 | 135,289.47 | 6.37 | 53.39 | −0.653 | Chloroplast. Mitochondrion. Nucleus |
| ZmFAR02.2 | GRMZM2G463730 | 1,202 | 135,849.03 | 6.14 | 52.73 | −0.664 | Chloroplast. Mitochondrion |
| ZmFAR03.1 | GRMZM2G155980 | 863 | 97,236.72 | 5.76 | 41.19 | −0.356 | Nucleus |
| ZmFAR03.2 | GRMZM2G155980 | 870 | 98,009.58 | 5.81 | 40.98 | −0.365 | Chloroplast. Nucleus. |
| ZmFAR04.1 | GRMZM2G034868 | 817 | 94,179.37 | 5.31 | 47.83 | −0.665 | Nucleus |
| ZmFAR04.2 | GRMZM2G034868 | 772 | 89,257.18 | 5.46 | 47.97 | −0.614 | Nucleus |
| ZmFAR05.1 | GRMZM2G302323 | 850 | 96,714.61 | 6.77 | 51.62 | −0.427 | Chloroplast. Nucleus. |
| ZmFAR06.1 | GRMZM2G043250 | 687 | 78,325.27 | 6.26 | 39.68 | −0.572 | Nucleus |
| ZmFAR07.1 | GRMZM2G117108 | 879 | 100,616.89 | 6.08 | 45.16 | −0.573 | Chloroplast |
| ZmFAR07.2 | GRMZM2G117108 | 883 | 101,135.51 | 6.08 | 44.66 | −0.567 | Chloroplast |
| ZmFAR08.1 | GRMZM2G106653 | 805 | 91,773.36 | 8.99 | 55.30 | −0.383 | Chloroplast |
| ZmFAR09.1 | GRMZM2G406651 | 855 | 94,584.69 | 7.02 | 52.79 | −0.378 | Chloroplast |
| ZmFAR09.2 | GRMZM2G406651 | 855 | 94,584.69 | 7.02 | 52.79 | −0.378 | Chloroplast |
| ZmFAR10.1 | GRMZM2G048987 | 690 | 79,156.85 | 6.29 | 40.42 | −0.634 | Chloroplast. Nucleus. Vacuole |
| ZmFAR10.2 | GRMZM2G048987 | 779 | 88,945.87 | 6.47 | 44.02 | −0.631 | Cell wall |
| ZmFAR10.3 | GRMZM2G048987 | 779 | 88,945.87 | 6.47 | 44.02 | −0.631 | Cell wall |
| ZmFAR11.1 | GRMZM2G129311 | 956 | 107,796.18 | 6.15 | 42.91 | −0.431 | Cell wall. Nucleus |
| ZmFAR11.2 | GRMZM2G129311 | 899 | 101,760.36 | 6.35 | 44.47 | −0.455 | Cell wall. Nucleus |
| ZmFAR12.1 | GRMZM2G114461 | 704 | 80,279.66 | 7.79 | 48.05 | −0.434 | Nucleus |
| ZmFAR12.2 | GRMZM2G114461 | 648 | 74,258.86 | 8.50 | 47.52 | −0.435 | Nucleus |
| ZmFAR13.1 | GRMZM2G148940 | 673 | 76,583.46 | 8.89 | 43.22 | −0.329 | Chloroplast. Nucleus |
| ZmFAR14.1 | GRMZM2G104268 | 1,023 | 115,820.99 | 6.03 | 48.53 | −0.408 | Nucleus |
| ZmFAR14.2 | GRMZM2G104268 | 1,047 | 118,413.75 | 5.94 | 48.49 | −0.414 | Nucleus |

**Note:**
Protein characterization information (Number of amino acids, Molecular weight, Theoretical pI, Instability index, Grand average of hydropathicity) was obtained from the ProtParam (https://web.expasy.org/protparam/). Plant-mPLoc (http://www.csbio.sjtu.edu.cn/) was used for subcellular localization analysis.

hydrophilic proteins. The instability index of ZmFAR06.1 was 39.68, which was the only stable protein in the ZmFAR family, while the rest members with instability indexes greater than 40, which belonged to unstable proteins. Plant-mPLoc localization analysis revealed that ZmFAR proteins were not only found in the nucleus, but also distributed in chloroplast, mitochondrion, vacuole, and cell wall, suggesting the evolution of potentially new functions in these locations for these proteins.

## The secondary structure and conserved structural domains analysis of ZmFARs proteins

The secondary structure of ZmFARs were analyzed by SOMPA online software. The results showed that all ZmFAR members contained four conformations, with the highest proportion of α-helical and random coil structures, accounting for more than 80%.

The second highest proportion is extended strand, accounting for 11.46–15.32%. And the lowest proportion is β-turned structures, accounting for only 3.00–6.84%. All proteins do not contain a beta sheet structure (Table S5).

The conserved motifs analysis of ZmFAR members were performed using MEME online software. Eight conserved motifs were set for testing. The sequence of conserved motifs was shown in Fig. S5. Noticeably, motifs composition and arrangement were in good agreement with the phylogenetic tree (Fig. 2B). The ZmFAR01.1 protein of Group1 differed from the other ZmFAR family proteins greatly, and it contained only Motif4 and Motif7, indicating that the C-terminal of ZmFAR01.1 is very different from other proteins, which is consistent with the result of MegAlign comparison of protein sequences (Fig. S2). The motifs position and number of the remaining ZmFAR proteins were highly conserved and similar, indicating these genes might have similar biological functions. However, ZmFAR09.1 and ZmFAR09.2 in Group2 did not contain Motif7. Members in Group4 did not contain Motif7 except for ZmFAR14.1 and ZmFAR14.2 (Fig. 2B), suggesting *ZmFAR14* may have different biological functions from *ZmFAR3*, *ZmFAR8*, and *ZmFAR7*.

## The genes structures and promoter analysis of *ZmFARs*

Analyzing gene structure, especially the distribution and number of introns and exons, is very important for studying gene's function. Therefore, we investigated the genes structures of ZmFHY3/FAR1 members. The results showed that *ZmFAR01.1* in Group1 contained 4 introns; members in Group2 contained a large variation in the number of introns, containing 1–7 introns; Group3 had no introns; members in Group4 contained 4–10 introns; and members in Group5 contained 0–3 introns (Fig. 2D). UTR plays important roles in gene regulation and mRNA stability (*Barrett, Fletcher & Wilton, 2012*). The *ZmFAR08.1* gene did not contain UTR, and the rest of the *ZmFARs* contained either 5'UTR or 3'UTR (Fig. 2D). It can be seen that the gene structures of the subfamilies are significantly different. Interestingly, ZmFAR family members in the same group revealed a high degree of similarity in the arrangement and distribution of exons, indicating that they might have similar biological functions. It is worth noting that nine of the fourteen genes have alternative splicing forms, and they encode at least two variations, indicating *ZmFAR1s* may function through variable splicing to increase the functional complexity of genes under certain conditions.

*Cis*-acting elements in the promoter often determine the expression and function of genes (*Hernandez-Garcia & Finer, 2014*). In order to explore the *ZmFARs* expression patterns, we analyzed its promoter by PlantCare website. There are abundant tissue-specific expression response elements, stress response elements, and hormone response elements in the promoter regions (Fig. 5). The stress response elements included high temperature (STRE), low temperature (LTR), temperature (TCA), drought (CCAAT-box, DRE core, DRE1, MBS, MRE, Myb, MYB recognition site, Myb-binding site, MYB-like sequence), anaerobic (ARE and GC-motif), damage (WRE3, W box and WUN-motif), defense and stress response (TC-rich repeats), *etc*. Noticeably, the drought stress response elements contained nine kinds. Hormone response elements included gibberellins (P-box, TATC-box, and GARE-motif), jasmonic acid (MYC, Myc, and JERE),

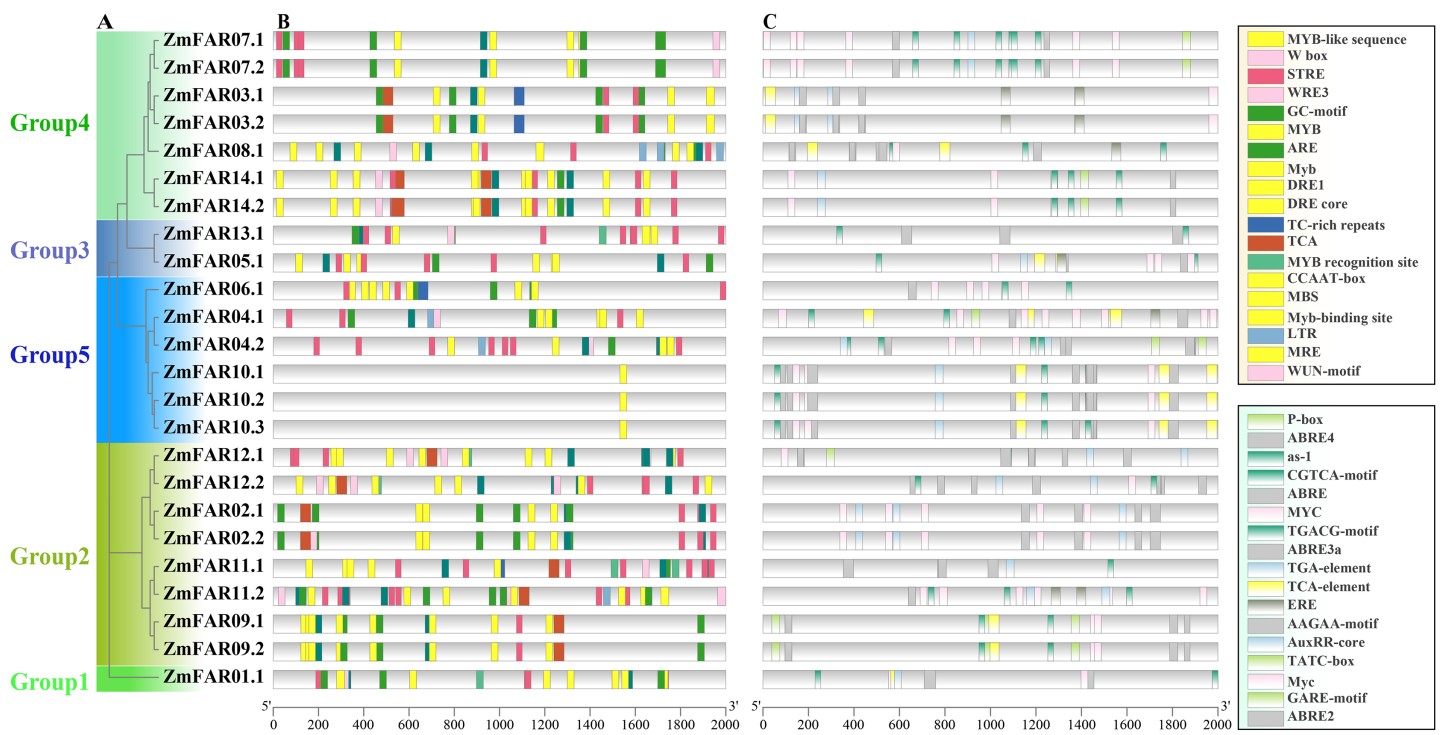

**Figure 5 Cis-acting elements in promoters of ZmFARs.** (A) The unrooted phylogenetic tree of ZmFARs. (B) *Cis*-elements on promoters associated with stress response. (C) *Cis*-elements on promoters associated with hormone response. Boxes with different colors represent different *cis*-element identified by the PlantCARE, and each colored box representing a different motif, shown in the right.

methyl jasmonate (CGTCA-motif, TGACG-motif, and as-1), auxin (TGA-element, TGA-box, and AuxRR-core) salicylic acid (TCA-element, and SARE), abscisic acid (AAGAA-motif, ABRE, ABRE2, ABRE3a, and ABRE4), ethylene (ERE), and others. Especially, ABA response elements are abundant in type and number. The presence of these elements on the promoter implies that the *ZmFARs* may be involved in abiotic stress response in maize.

## Analysis of tissue expression patterns of *ZmFARs*

In order to elucidate the expression patterns of *ZmFARs* in various tissue during maize development, we analyzed the transcriptome data downloaded from the NCBI. As shown in Fig. 6, *ZmFARs* displayed different expression patterns in different tissues. Most *ZmFARs* were highly expressed in the primordium, germinatin kernel, embryo, meristem, leaf, and internode. However, the expression levels in mature pollen, silk, root, endosperm and pericarp/aleurone were low. The expression of *ZmFARs* varied in different tissues, with *ZmFAR01* in Group1 being the most different from the other members. The highest expression of *ZmFAR01* was found in mature leaf 8, while all other *ZmFARs* had lower expression. This result indicated that *ZmFAR01* played more important roles in mature leaf than the other genes. The expression levels of ZmFAR members in Group5 were high at all tissues, and especially *ZmFAR04* was most highly expressed in the primordium. The above results suggested that *ZmFARs* may play vital roles in different tissues during maize development.

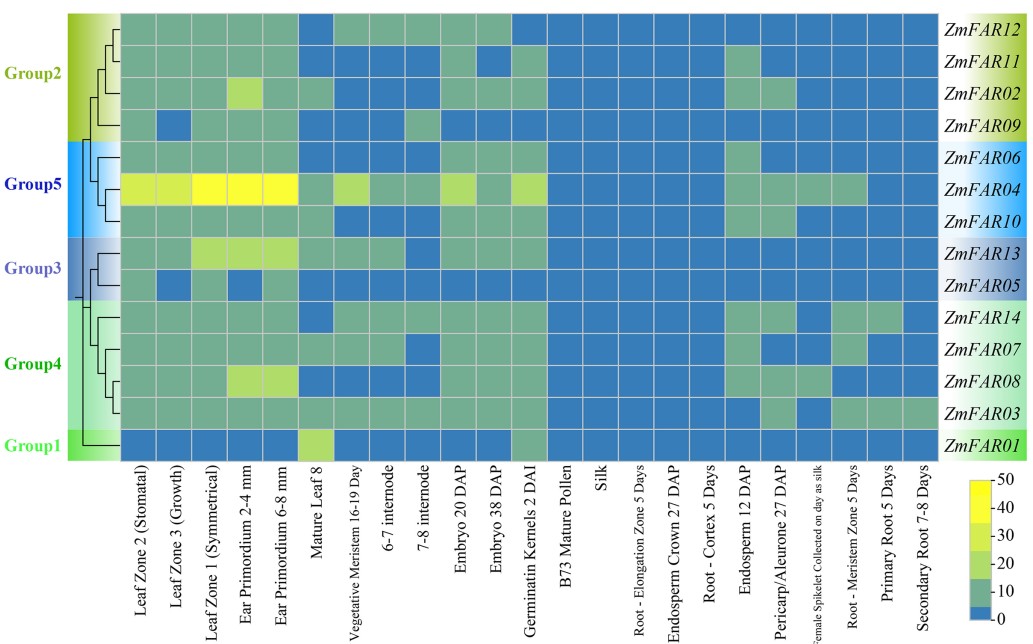

**Figure 6** The expression profiles of *ZmFARs* in various tissues of maize growth stage.

## Expression analysis of *ZmFARs* under drought stress

The *ZmFARs* promoter contains many drought stress response elements, so we suppose that the *ZmFARs* might be involved in drought stress response in maize. At the early seedling stage, root system and mesocotyls are critical for the early vigour of maize seedlings (*Peter et al., 2009*; *Saenz Rodriguez & Cassab, 2021*). The expression level of *ZmFARs* in primary roots, seminal roots, lateral roots and mesocotyls of B73 seedlings before and after drought stress treatment were detect by qRT-PCR. The results showed that most *ZmFARs* were down-regulated in root and mesocotyls after drought stress (Fig. 7). *ZmFAR01* was significantly down-regulated in primary roots, lateral roots and mesocotyls, and its expression level decrease about 2/3 under drought stress than under normal conditions (Fig. 7A). *ZmFAR04*, *ZmFAR13*, and *ZmFAR14* were significantly down-regulated in primary root, seminal root, and mesocotyl, but the fold change was not large (Figs. 7D, 7M, and 7N). *ZmFAR02*, and *ZmFAR05* were significantly down-regulated in mesocotyl (Figs. 7B, and 7E). *ZmFAR03*, *ZmFAR08*, and *ZmFAR09* were significantly down-regulated in primary root (Figs. 7C, 7H, and 7I). *ZmFAR07* were significantly down-regulated in primary root and seminal root (Fig. 7G), while *ZmFAR10* down-regulated in primary root and lateral root (Fig. 7J). *ZmFAR06*, *ZmFAR11*, and *ZmFAR12* were down-regulated in mesocotyl after drought stress (Figs. 7F, 7K, and 7L).

As a non-permeable, non-ionic osmoticum, polyethyleneglycol (PEG) 6000 cannot enter the pores of plant cell wall space (*Oertli, 1985*), and many early studies also used PEG-6000 solution to induce drought stress (*Liu et al., 2024*; *Suslov, Daminova & Egorov, 2024*). It is a better choice for imposing low water potential, causing a drought stress. To further investigate the expression of *ZmFARs* in maize roots at the three-leaf stage

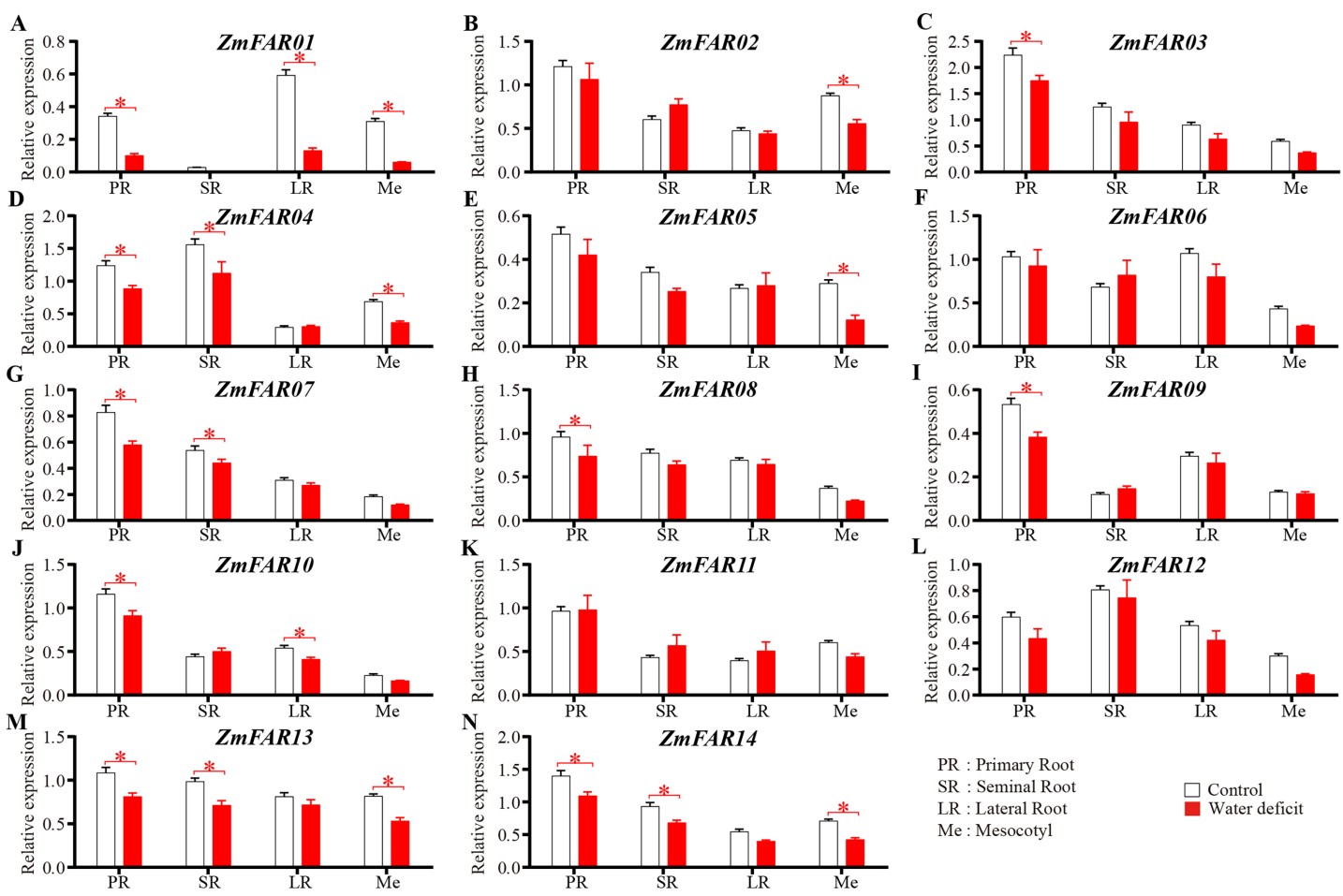

**Figure 7  At the early seedling stage, the expression patterns of *ZmFARs* in root system and mesocotyls under water deficit.** The bars indicate the mean ± SD of three replicates (Student's *t* test, *$p < 0.05$).

under drought conditions, 10% (w/v) and 25% (w/v) PEG-6000 aqueous solution were used to treat B73 seedlings. qRT-PCR was performed to detect the expression of *ZmFARs* after 6, 24, and 48 h of treatment, respectively. The results showed that the expression of *ZmFARs* were mostly down-regulated in root after PEG-6000 treatment (Fig. 8), which was consistent with the expression patterns in root at the early seedling stage. Compared with control, the expression patterns of *ZmFAR01*, *ZmFAR03*, *ZmFAR05*, *ZmFAR06*, *ZmFAR09*, *ZmFAR12*, *ZmFAR13*, and *ZmFAR14* showed a continuous decrease tendency from 6 to 48 h treatments. All of them were significantly down-regulated, but the fold changes were slightly different. The remaining *ZmFARs* showed inconsistent expression trends under different concentrations of PEG-6000 treatments. For example, after 10% PEG treatment, the expression of *ZmFAR02* showed a decreasing trend after 6 h, an increasing trend after 24 h, and a decreasing trend after 48 h. Whereas, *ZmFAR02* was significantly down-regulated after 25% PEG treatment for all time detected (Fig. 8B). The expression of *ZmFAR04* and *ZmFAR10* showed up-regulation after 10% PEG-6000 treatment, but not significantly (Figs. 8D and 8J). These results suggested that *ZmFARs*

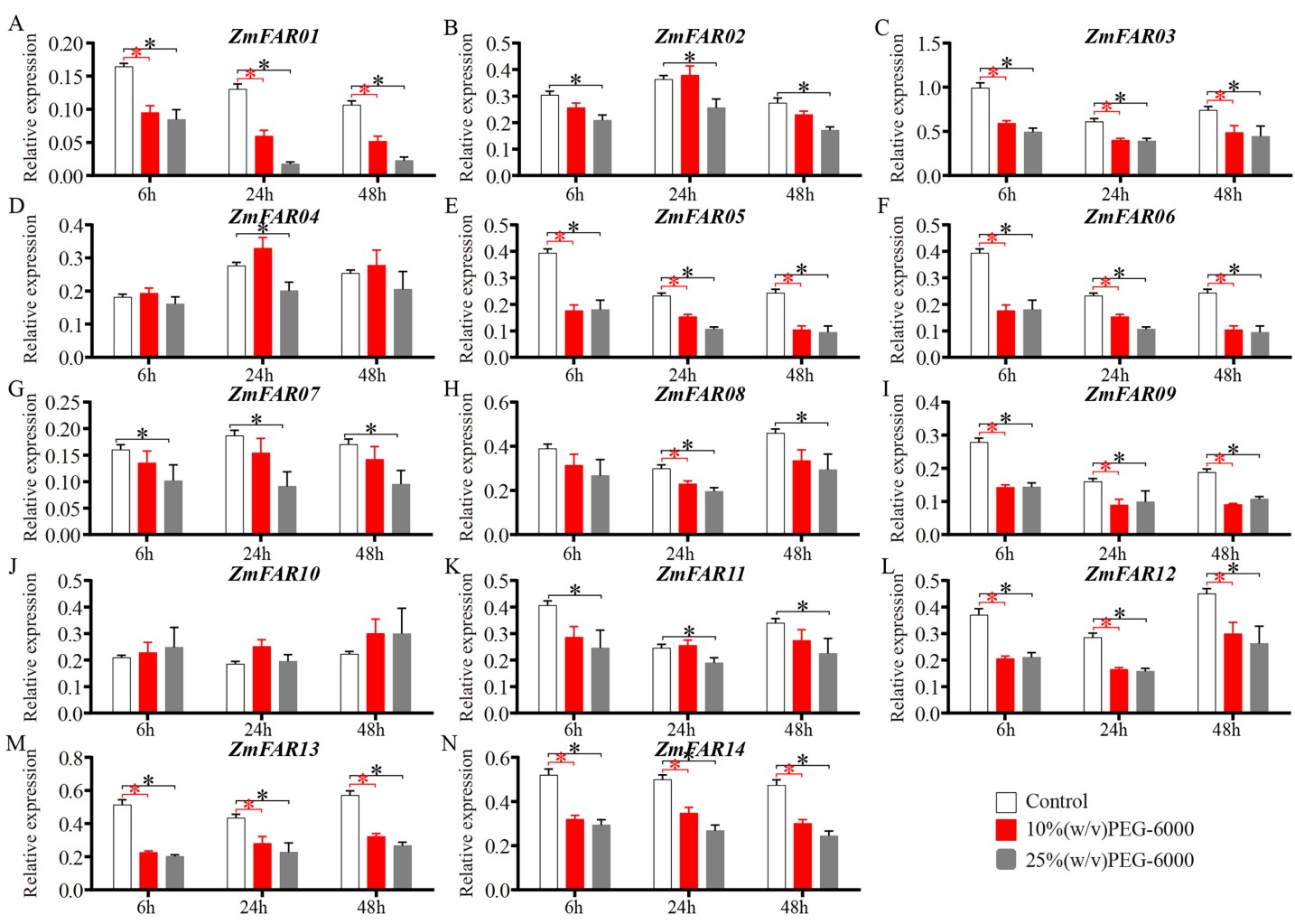

**Figure 8 The expression patterns of *ZmFARs* in root under different concentrations of PEG-6000 treatment.** Seedlings with no treatment as control. The bars indicate the mean ± SD of three replicates (Student's *t* test, $^*p < 0.05$).

may be functional redundancy and play negatively roles in the regulation of target genes expression by altering its own expression in root of maize under drought stress conditions.

## DISCUSSION

In this study, we identified 24 ZmFAR members encoded by 14 genes (Fig. 1), which all contain the FHY3 structural domain (Fig. 2B). FAR1/FHY3 proteins contain three functional structural domains, including the C2H2-type zinc finger domain with DNA-binding activity, the putative core transposase domain, and the SWIM zinc finger domain with transcriptional activation activity (*Lin et al., 2008*). All ZmFAR members have conserved CCHH motif in N-terminal (Fig. S2). The "DDE" catalytic triad motif in the middle position, as well as the SWIM zinc-finger domain at the C-terminal with conserved CxCxnCXH motif, are essential for their homodimerization or heterodimerization with other FAR1 (*Lin et al., 2008*). Evolutionary relationships and conserved motifs analyses revealed that the protein sequence of ZmFAR01 differs

significantly from the other ZmFARs (Figs. 2 and 3), indicating ZmFAR01 may function differently from the other proteins, although its expression pattern is similar to that of the other genes after drought treatment in root (Figs. 7 and 8). Compared to other tissues, *ZmFAR01* is highly expressed in mature leaves, suggesting that *ZmFAR01* functions mainly in mature leaves. But what it functions? To regulate leaf response to light, to regulate leaf growth and development (leaf senescence), to regulate chloroplast synthesis in leaves, or to regulate leaf response to drought stress similar to the function in roots, all of which need to be verified.

FAR1/FHY3 play multiple roles in a wide range of cellular processes, including light signal transduction, photomorphogenesis, regulation of the biological clock and flowering time, stem meristem and flower development, chloroplast division, chlorophyll biosynthesis, starch synthesis, abscisic acid response, oxidative stress response, plant immunity, and low-phosphorus response, *etc* (*Ma & Li, 2018*). Many transcription factors, proteins involved in cell wall extension, and related to redox balance control reduced responsiveness in *fhy3* (*Hudson, Lisch & Quail, 2003*). Subcellular localization analysis showed that ZmFARs were distributed in nucleus, chloroplast, mitochondrion, vacuole, and cell wall (Table 1). In Arabidopsis, *FRS1*, *FRS8*, and *FRS9* are targeted into the nucleus despite their lack of predicted NLSs (*Lin & Wang, 2004*). In tea plants, *CsFRS1*, *CsFRS7*, *CsFRS14*, *CsFRS16*, *CsFRS18*, and *CsFRS23* are predicted to target other cellular components besides the nucleus (*Liu et al., 2021*). This suggested that ZmFAR1s might enter the nucleus by interacting with other members of ZmFAR1s to form a homologous or heterodimers to regulate target genes expression under certain specific conditions. But the subcellular localization of ZmFAR1 needs further experimental verification.

Previous reports FAR1 family members exhibit different tissue-specific expression patterns in species. For example, in Arabidopsis, the *AtFAR1s* were expressed in leaves, stems, and flowers (*Lin & Wang, 2004*). In cotton, most genes were highly expressed in leaves and only a few genes were expressed in the stem, petal, and torus (*Yuan et al., 2018*). About 36.2% of *AhFAR1s* specific expression in flower, peg, leaf, root, and stem, 34.1% of genes were specifically expressed in shells and seeds (*Lu et al., 2022*). We found that most of the *ZmFARs* were expressed in high levels at sites of vigorous growth, such as primordium, germinatin kernel, embryo, meristem, leaf, and internode (Fig. 6), suggesting that *ZmFARs* plays an important role in regulating the growth and development of maize.

Improvement of root system architecture has been the goal of modern breeding programs to produce drought-tolerant varieties (*Ranjan et al., 2022*). Many transcription factors families have been identified participating in gene expression regulation or having an impact on root development under drought stress conditions (*Janiak, Kwasniewski & Szarejko, 2016*). The enrichment of the *ZmFARs* promoter regions with drought stress-responsive *cis*-acting elements predicts that the ZmFAR family members may be involved in drought stress response in maize. At early growth and development of B73 seedlings, most of the *ZmFARs* were down-regulated in primary roots, lateral roots and mesocotyls (Fig. 7). This is consistent with the results of different concentration of PEG simulated drought treatment of B73 seedlings in root at the three-leaf stage (Fig. 8). However, some *ZmFARs* were also up-regulated, such as *ZmFAR04* and *ZmFAR10*.

In barley, some *HvFRF* genes were significantly upregulated in response to drought stress, and *HvFRF9* overexpression could enhance drought resistance in *Arabidopsis* (*He et al., 2024*). *MYB96* reduces lateral root growth and enhances drought tolerance in plants by integrating ABA and auxin pathways (*Seo et al., 2009*). It has been found that root growth was not or only slightly affected at −0.2 MPa for short-term (*Sharp et al., 2004*), and lower water potentials −0.8 MPa can significantly reduce primary root elongation (*Opitz et al., 2014*; *Sharp et al., 2004*). Both concentrations of 10% (about −0.15 MPa) and 25% PEG-6000 (about −0.15 MPa) taken in this study to simulate drought (*Michel & Kaufmann, 1973*), root growth was affected and root hairs were significantly reduced. After 48 h of treatment, partial necrosis had appeared at the root tip site (Fig. S1). There is an epistatic role of FAR1/FHY3 family genes, which regulate the expression of other transcription factors and essential for plant growth and development (*Li et al., 2016*; *Lin et al., 2007*). We speculate that ZmFARs regulates plant drought stress resistance, may integrate ABA signalling and may regulate ABI5. It is also possible that ROS signalling may be integrated. This conjecture needs to be subsequently verified. In addition to involvement in drought, the ZmFAR1 family also has potential applications in inflorescence development (*Tang et al., 2024*).

## CONCLUSIONS

In short, we performed a systematic identification and analysis of the FAR1/FHY3 family genes in maize. A total of twenty-four ZmFAR members, named ZmFAR01-14, were identified in the maize through a genome-wide study. ZmFARs can be divided into five subgroups based on their phylogenic relationships, and the protein and gene structures of each subfamily varied greatly. The promoter regions of *ZmFARs* contained abundant stress-responsive and hormone-responsive *cis*-elements, especially drought-responsive *cis*-elements. *ZmFARs* were expressed in all tissues of maize, but the expression level varies greatly. Most *ZmFARs* were down-regulated in primary roots, seminal roots, lateral roots and mesocotyls of maize under drought stress, implying that the FAR1/FHY3 family has important roles in plant growth and development, and drought stress response. FAR1/FHY3 family may negatively regulate drought stress resistance in maize. These results lay the foundation for analysis of the functions of *ZmFARs* in response to abiotic stresses, and also provide potential genetic resources for the genetic improvement of drought-tolerant maize lines.

### Funding

This work was supported by the Agricultural Variety Improvement Project of Shandong Province (No. 2022LZGC019-3); the Shandong Agricultural Industry Technology System (No. SDAIT-02-19); the China Agriculture Research System (No. CARS-02); National Natural Science Foundation of China (No. 32201714); the Shandong Province Higher Educational Science and Technology Program (No. 2023KJ270); and the Dezhou University Innovation Team Program (No. DZUQC202301). The funders had no role in

study design, data collection and analysis, decision to publish, or preparation of the manuscript.

## Grant Disclosures

The following grant information was disclosed by the authors:
Agricultural Variety Improvement Project of Shandong Province: 2022LZGC019-3.
Shandong Agricultural Industry Technology System: SDAIT-02-19.
China Agriculture Research System: CARS-02.
National Natural Science Foundation of China: 32201714.
Shandong Province Higher Educational Science and Technology Program: 2023KJ270.
Dezhou University Innovation Team Program: DZUQC202301.

## Competing Interests

The authors declare that they have no competing interests.

## Author Contributions

- Dongbo Zhao conceived and designed the experiments, performed the experiments, analyzed the data, prepared figures and/or tables, authored or reviewed drafts of the article, and approved the final draft.
- Peiyan Guan performed the experiments, analyzed the data, prepared figures and/or tables, authored or reviewed drafts of the article, and approved the final draft.
- Longxue Wei performed the experiments, analyzed the data, prepared figures and/or tables, and approved the final draft.
- Jiansheng Gao analyzed the data, prepared figures and/or tables, and approved the final draft.
- Lianghai Guo analyzed the data, prepared figures and/or tables, and approved the final draft.
- Dianbin Tian analyzed the data, prepared figures and/or tables, and approved the final draft.
- Qingfang Li analyzed the data, prepared figures and/or tables, and approved the final draft.
- Zhihui Guo analyzed the data, prepared figures and/or tables, and approved the final draft.
- Huini Cui analyzed the data, prepared figures and/or tables, and approved the final draft.
- Yongjun Li analyzed the data, prepared figures and/or tables, and approved the final draft.
- Jianjun Guo conceived and designed the experiments, analyzed the data, prepared figures and/or tables, authored or reviewed drafts of the article, and approved the final draft.

## Data Availability

   Raw data are available in the Supplemental Files.
## Supplemental Information

Supplemental information for this article can be found online at http://dx.doi.org/10.7717/peerj.17684#supplemental-information.

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
