# Peer review of "Comprehensive identification and expression analysis of FAR1/FHY3 genes under drought stress in maize (Zea mays L.)"

_PeerJ, doi:10.7717/peerj.17684_

## Round 0.1 · original submission · Minor Revisions

Overall, your manuscript seems well constructed and written. Acceptable after careful minor revision. Reviewers made valid criticisms. Here are a few minor points I would like to add.

-I do not find the interparagraph relocation suggested by reviewer 2 for the introduction section appropriate. I think it's better the way it is.
-Sugarcane and Arobidopsis are okay, but I think it would be more appropriate to exclude tomatoes and potatoes from the evolutionary relationships data. I suggest you remove their findings and redo the analysis.
-"Drought is a major threat to global crop production, affecting human health and crop productivity for about one third of the world's population (Sheoran et al. 2022)." The sentence is unnecessary. It should be deleted.

·

Basic reporting

The manuscript was interesting and impactful for maize researchers to further study of FHY3/FAR1 family under drought stress in maize

Experimental design

The approach and analysis of the study are sound

Validity of the findings

The result and discussion part adequate

Additional comments

Missing supplementary table and figures caption, should be added after conclusion.
Fig. 1: Should be show the physical position of each gene, that will be more informative for researcher.

Reviewer 2 ·

Basic reporting

Global identification and expression analysis of FAR1 FHY3 genes under drought stress in maize (Zea mays L).
I think the title needs to be rephrased. Also, 'methods' should be written as 'method' in the abstract.
Additionally, the method section in the abstract is fragile.
Lines 90-109 should be the first passage of the introduction section, and this should be followed by Lines 76-89.

Experimental design

Plant material and treatment are quite complicated. Also, please simplify it so that international English speakers can understand. This section is insufficient.

Validity of the findings

Please remove Chr.02, Chr06, Chr08 from Figure 1.
Line285-296. Only some gene results are written, where are the remains genes, please give information related to them.
In the discussion section, Lines 317-327. This section is not required. Directly should start with Line 328 or use the 367-374 as the first passage of the discussion section.
The manuscript is almost 85% written with data downloaded from the internet. Only a small portion relies on analysis results. There are no references to online or computer package programs. The Discussion section is filled with numerous irrelevant citations that have no connection to the results. The Discussion section is supposed to discuss reasons for similarities or differences with other studies, but unfortunately, such discussion is lacking here. The Conclusion section serves as a summary of the article. I believe this article, which is primarily based on analyses of downloaded data from the internet, needs to be revised extensively. A detailed explanation of the application of PEG-6000 is required
There is no information about the information source of Table 1.

Reviewer 3 ·

Basic reporting

The manuscript was written in clear and unambiguous professional English. Sufficient field background and literature references were given. The manuscript has good structure consisting of figures and tables.

Experimental design

There are two concerns:

1. For Section 3.3 Collinearity analysis of ZmFARs, I recommend to add a brief description in the paragraph to indicate why collinearity analysis is needed and what software was used to get the collinearity analysis results.

2. In Section 3.8 Expression analysis of ZmFARs under drought stress, please clarify what software was used to obtain the statistical analysis results. Also indicate the statistical methods you used for two group comparison or multiple group comparison to get Figure 7 and 8. For figure 8, did the author used raw p-value or adjusted p-value to control the family-wise error rate existing in multiple group comparison?

Validity of the findings

FAR1/FHY3 transcription factors, originating from transposases, regulate gene expression in various processes, including light signal transduction, growth, development, and stress response. Despite their identification in multiple species, their function in maize, particularly in drought, remains unclear. The authors conducted a comprehensive analysis of maize FAR1 genes, identifying 24 ZmFAR members grouped into 5 subfamilies. Promoter regions of ZmFARs contain abundant drought-responsive elements. Expression analysis suggests a potential role for FAR1/FHY3 in maize drought response, warranting further investigation. However, the concerns need to be resolved so that the findings of this manuscript are promising.

Additional comments

NA

·

Basic reporting

The MS involves a genome-wide and systematic identification of the FAR1/FHY3 genes in maize and reported that most ZmFARs were down-regulated in root after drought stress indicating that FAR1/FHY3 transcription factors have important roles in drought stress response. The topic is recent and the results add important information to understanding how maize plants can resist drought conditions.
However, the authors should consider the following comments to improve the presentation of the manuscript
1) The literature includes too many examples of dicot plants, it should have included more examples from the monocots, especially the cereals. The literature on the examined genes is too excessive and could have been focused on the most relevant plants to maize. This is confirmed by the lack of orthologous genes in maize (Fig. S3), and dicots (Fig. S4) and the presence of of 14, 12, 11, 16, and 8 paralogous genes in maize and the monocot plants Sorghum bicolor, Oryza sativa, Oryza indica, Saccharum spontaneum, and Hordeum vulgare, respectively. Excessive literature is not necessary to justify the performed study. See also comments No. 3.
2) The authors used abbreviations for scientific names of some plants, whereas a full name is more informative like using A. hypogaea in line 81 of the text
3) I have reservations about the evolutionary relationships of FAR1/FHY3 transcription factors in maize, sugarcane, Arabidopsis, tomato, and potato. It seems of little value since it did not differentiate the FAR1/FHY3 of zea maize (red dots) from those of other species. Since these plants are taxonomically unrelated, the use of their FAR1/FHY3 to construct a phylogenetic tree is not relevant to the topic of the MS which is “identiûcation and expression analysis of FAR1/FHY3 genes under drought stress in maize”. It merely indicates that these genes evolved early in the evolution of the angiosperms. This figure may be omitted
4) The discussion also included excessive reference to unnecessary literature especially the paragraph starting from “line 367 to line 374”. It adds no insights to the topics and can also be deleted. Stating that “ Drought is a major threat to global crop production, affecting human health and crop productivity for about one-third of the world’s population and that drought stress-related genes are an effective strategy to improve crop drought resistance is well-known information said many times. Again the examples of Arabidopsis and peanuts were stated in the Introduction and have little relevance to maize. See comments 1 and 3.
5) The authors concluded that their results can lay the foundation for further analysis of the functions of ZmFARs in response to abiotic stresses, and also provide potential genetic resources for the genetic improvement of drought-tolerant maize lines. The author's conclusion should focus on how these results can lay a foundation for the functions of ZmFARs in response to drought stress not lying grounds for further research.

Experimental design

Original primary research within the Aims and Scope of the journal.
The research question is well-defined, relevant & meaningful. It is stated how research fills an identified knowledge gap.
Rigorous investigation performed to a high technical & ethical standard.
Methods described with sufficient detail & information to replicate.

See basic reporting

Validity of the findings

All underlying data have been provided; they are robust, statistically sound, & controlled.

However, the conclusions are well stated and linked to the original research question but are not limited to supporting results.
The findings are valid and mostly acceptable, See basic reporting

Additional comments

See basic reporting

---

## Round 0.2 · accepted · Accept

Dear Authors

Because you have improved your manuscript by taking into account reviewers' and editorial comments, I am pleased to report that it has been accepted.
Congratulations.

·

Basic reporting

All comment addressed.

Experimental design

All comment addressed.

Validity of the findings

All comment addressed.

Additional comments

It's pleased to notify you that the author has addressed all of the proposed comments

Reviewer 2 ·

Basic reporting

From my side it is ok. the author did the major corrects.

Experimental design

ok

Validity of the findings

ok

Additional comments

ok

Reviewer 3 ·

Basic reporting

The revised manuscript was written using clear and unambiguous professional English. The literature references, background information are sufficient. The structure, figures and tables look good.

Experimental design

The revised manuscript followed the guidance and comments of reviewers. The experimental design makes sense. Research questions were well defined and meaningful. The investigation is rigorous. Methods were well described and sufficient.

Validity of the findings

Since the experimental design is reasonable, the results and conclusions are promising.

Additional comments

NA